# Rheumatoid Arthritis Associated Episcleritis and Scleritis: An Update on Treatment Perspectives

**DOI:** 10.3390/jcm10102118

**Published:** 2021-05-14

**Authors:** Veronique Promelle, Vincent Goeb, Julie Gueudry

**Affiliations:** 1Department of Ophthalmology and Visual Sciences, The Hospital for Sick Children, University of Toronto, Toronto, ON M5G 1X8, Canada; 2EA 7516 CHIMERE, Université de Picardie Jules Verne, 80025 Amiens, France; goeb.vincent@chu-amiens.fr; 3Department of Rheumatology, Centre Hospitalier Universitaire Amiens Picardie, 80054 Amiens, France; 4Department of Ophthalmology, Hospital Charles Nicolle, 76000 Rouen, France; julie.gueudry@chu-rouen.fr; 5EA7510, UFR Santé, Rouen University, F-76000 Rouen, France

**Keywords:** scleritis, episcleritis, rheumatoid arthritis, ocular inflammation, biologics, treatment, prognosis

## Abstract

Episcleritis and scleritis are the most common ocular inflammatory manifestation of rheumatoid arthritis. Rheumatoid arthritis (RA) accounts for 8% to 15% of the cases of scleritis, and 2% of patients with RA will develop scleritis. These patients are more likely to present with diffuse or necrotizing forms of scleritis and have an increased risk of ocular complications and refractory scleral inflammation. In this review we provide an overview of diagnosis and management of rheumatoid arthritis-associated episcleritis and scleritis with a focus on recent treatment perspectives. Episcleritis is usually benign and treated with oral non-steroidal anti- inflammatory drugs (NSAIDs) and/or topical steroids. Treatment of scleritis will classically include oral NSAIDs and steroids but may require disease-modifying anti-rheumatic drugs (DMARDs). In refractory cases, treatment with anti TNF biologic agents (infliximab, and adalimumab) is now recommended. Evidence suggests that rituximab may be an effective option, and further studies are needed to investigate the potential role of gevokizumab, tocilizumab, abatacept, tofacitinib, or ACTH gel. A close cooperation is needed between the rheumatology or internal medicine specialist and the ophthalmologist, especially when scleritis may be the first indicator of an underlying rheumatoid vasculitis.

## 1. Introduction

Rheumatoid arthritis (RA) is a systemic inflammatory disorder affecting the joints, which can have several extra-articular manifestations, including inflammatory ocular disease. Ocular manifestations, including episcleritis, scleritis, peripheral ulcerative keratitis, and dry eye disease, can be encountered in as many as 39% of patients with RA [1]. Patients with positive anti-cyclic citrullinated peptide (anti-CCP) antibody or, even more so, patients with both positive anti-CCP antibodies and positive rheumatoid factor seem to be more likely to have some ocular involvement than other patients [1,2]. Apart from dry eye disease, episcleritis and scleritis are the most common, accounting for 25% and 36%, respectively, of patients with inflammatory ocular manifestation associated with rheumatoid arthritis [3]. It was estimated that approximately 5% of patients with RA will develop episcleritis, and 2% will develop scleritis [1,4,5]. Peripheral ulcerative keratitis (PUK), another sight-threatening ocular manifestation of RA, occurs in 1 to 2% of patients with RA [1,4] and accounts for 23% of patients with RA-associated inflammatory ocular disease [3].

The management of scleritis in rheumatoid arthritis can be challenging, and in severe cases lead to the loss of vision, even to the loss of the eye. Fortunately, more and more immunomodulatory medications are developed for the treatment of RA, that seem effective in controlling the refractory inflammation in scleritis. Because of their knowledge of inflammatory disorders and immunosuppressive medications, rheumatology and internal medicine specialists may be asked for assistance in the diagnosis and management of ocular inflammation, associated with RA or not.

In this review we aimed to provide an overview of recent data on the diagnosis and management of episcleritis and scleritis associated with rheumatoid arthritis, including an update on treatment perspectives.

## 2. Demographics, Presentation, and Classification of Episcleritis and Scleritis

A clinical classification was proposed in 1976 by Watson and Hayreh and has been widely adopted since [6]. In the normal anatomy of the outer vascular coats of the eye (Figure 1), the conjunctival plexus is the most superficial, a fine layer of vessels easily moving over the underlying structures. Underneath, the episcleral plexus has straight, radially arranged vessels. The deepest, scleral plexus has a pattern of criss-cross vessels tightly adherent to the sclera [6]. Episcleritis (Figure 2) can be *simple* or *nodular*. Anterior scleritis is classified into *nodular* (Figure 3), *diffuse* (Figure 4), *necrotizing with inflammation* (Figure 5), or *necrotizing without inflammation* (scleromalacia perforans) (Figure 6). In posterior scleritis (Figure 7), inflammation of the sclera involves the posterior part of the globe. Anterior scleritis is more common than posterior, and the most frequent forms are the nodular and diffuse forms. The majority of patients are female [3,5,6,7]. The median age at presentation varies between 51 and 69 years [3,7], patients with episcleritis presenting generally at a younger age than patients with scleritis.

Patients with anterior scleritis present with redness of the eye without discharge, and severe pain. The pain irradiates around the orbit, forehead and cheek, is usually worse during the night and may typically wake the patient at night [6]. The pain is most severe in necrotizing with inflammation form, but absent in necrotizing without inflammation. The visual acuity is usually preserved. By comparison, in posterior scleritis, the redness may be minimal or absent, and the patient will present with decreased vision related to serous retinal detachment (up to 40% of patients have a vision of less than 6/12 at presentation), and severe pain, rarely with proptosis and restriction of ocular movements [8]. In posterior scleritis, presentation is bilateral in 15% of cases, and associated anterior scleritis may be found in almost 20% of cases [8].

## 3. Episcleritis/Scleritis and Rheumatoid Arthritis

Scleritis is associated with a systemic condition in 18% to 35% of patients at the time of presentation. An additional proportion of patients will be diagnosed with a systemic disease later in the course of follow-up, so that at the end of follow-up 38% to 45% of patients with scleritis have a systemic diagnosis [7,9]. Among those, RA is the most common diagnosis, accounting for 8% to 15% of patients with scleritis [7,9,10,11].

Patients with RA are more likely to present with diffuse or necrotizing forms of scleritis [9,10] than patients with idiopathic scleritis, the latter more likely to present with nodular scleritis. No patient with posterior scleritis was found to have association with a systemic disease in the series of Yoshida et al. [10]. However, this could be only an effect of the rarer incidence of posterior forms, because in a series of 114 patients with posterior scleritis, Lavric et al. found an association with RA in 12% of cases [8].

It has been suggested that there could be a lower proportion of scleritis associated with systemic disease in Asian countries: in their series of 293 patients with scleritis in China, only 4% were found to have RA [12], which seems to be consistent with the findings that extra articular manifestations of RA are uncommon in southern Chinese populations [13].

In most cases, RA was diagnosed before the presentation of scleritis [10,11], as opposed to patients with ANCA-associated vasculitis such as granulomatosis with polyangiitis (GPA). In patients with idiopathic scleritis, the immunologic markers could be predictive of the risk of developing a systemic autoimmune disease. A positive rheumatoid factor (RF) was strongly associated with the risk of developing RA, while positive ANCA increased the risk of developing GPA. Presence of joint symptoms, type of scleritis, presence of ocular complications or positivity of ANCA, in patients with idiopathic scleritis, were not significant risk factors for developing RA during the course of follow-up [9].

In patients with RA-associated scleritis, Caimmi et al. found that the mean duration of RA disease before onset of scleritis was 15 years [3]. Episcleritis and scleritis were bilateral in approximately 30% of cases. More than 80% of patients had an auto-antibody positive disease (82% and 91% of episcleritis and scleritis patients, respectively). More importantly, the disease activity of RA at time of onset of ocular disease was absent or mild in 71% of scleritis and 41% of episcleritis patients. Radiographic erosions were present in 50 to 70% of patients. Approximately 35% of scleritis patients had presented rheumatoid nodules before the onset of inflammatory ocular disease. Most of the patients developed a scleritis or episcleritis while taking glucocorticoids and/or disease-modifying anti-rheumatic drugs (DMARDs), but 30 to 45% were treated with a biologic agent for the joint disease. Disease duration, articular disease severity, treatment or antibody profile seemed to be comparable in RA patients with ocular disease and matched patients without ocular disease [3].

Patients with RA or ANCA-related vasculitis are also more at risk of developing surgically-induced scleritis, a particular form of scleritis occurring in the aftermath of ocular surgery involving the sclera: typically strabismus surgery, scleral buckling, or pterygium surgery, or even after repeated intravitreal injections for age-related macular degeneration [11,14]. In those cases, more than ever, an effective cooperation between the ophthalmologist and the rheumatologist is crucial for early diagnosis and prompt treatment.

Lastly, it is important to keep in mind that infectious causes account for 5% of episcleritis and 13.5% of scleritis cases [6], secondary to tuberculosis, Syphilis, Herpes Simplex or Zoster virus mainly. Thorough investigation for infectious causes is mandatory for all cases of scleritis, especially in RA patients who can often be immunocompromised from their treatment. In the cases refractory to anti-inflammatory treatment, it is recommended to repeat and intensify the infectious work-up before escalating immunosuppressive treatment [15].

## 4. Treatment Modalities and Perspectives

### 4.1. Episcleritis

The management of episcleritis is most often fairly benign, involving oral non-steroidal anti-inflammatory drugs (NSAIDs) and/or topical steroids, topical NSAIDs and artificial tears [3,6]. In refractory or recurrent cases, periocular steroid injections, oral steroids, or, very rarely, DMARDs such as hydroxychloroquine, leflunomide, methotrexate, may be required [3].

### 4.2. Scleritis

Treatment strategies for scleritis classically include NSAIDs as a first line. Steroids, oral and/or intravenous pulse, may be required in more than 70% of patients [3]. Associated topical treatment can involve steroids, NSAIDs, cycloplegics, artificial tears, and/or cyclosporine [3].

DMARDs are reported to be used in half of the patients [3,10]: methotrexate [3,7,10,11], salazosulfapyridine [10], cyclophosphamide [3,7,11], cyclosporine A [7], or azathioprine [7]. To note, in a series of posterior scleritis, patients treated with mycophenolate mofetil (MMF) showed a significantly accelerated time of relapse compared to other patients [8].

### 4.3. Biologic Response Modifiers: Anti TNF Alpha

Treatment with biologics has been reported effective in small case series and case reports, with successful treatment of refractory cases with infliximab [16,17,18,19,20], adalimumab [21] or certolizumab pegol [22], then in larger series where up to 30% of scleritis patients required biologic response modifiers to achieve control of the inflammation [3,7,10,11]. In a retrospective series of 32 patients with ocular inflammation, Durrani et al. [23] included 9 patients with scleritis who were treated with adalimumab. Like in other features of ocular inflammation, etanercept seems to prove less effective in controlling the inflammation [22,24,25]. Infliximab has also been investigated in the treatment of scleritis in a phase I prospective open trial: all 5 participants, previously treated with oral prednisone only (*n* = 4) or prednisone and methotrexate (*n* = 1) achieved control of scleritis within 14 weeks of therapy [26]. Based on all the available evidence, recommendations from the American Academy of Ophthalmology were issued in 2014: anti-TNF therapy with infliximab or adalimumab should be considered for patients with scleritis who have failed first-line immunomodulatory therapies [27]. A randomized trial initiated in July 2019 will compare treatment with prednisone, infliximab, and low dose methotrexate for 16 weeks in one arm, to prednisone and cyclophosphamide in the other arm [ClinicalTrials.gov (accessed on 11 May 2021) Identifier: NCT03088293]. Thus, treatment with infliximab or adalimumab or cyclophosphamide should be proposed to patients who either were already on DMARD at the onset of scleritis or have failed a conventional DMARD, i.e., inability to control the inflammation, inability to decrease the steroids to an acceptable level, or multiple recurrences. Anti TNF alpha should also be considered as a first line agent in necrotizing scleritis with direct threat to vision (Figure 8).

### 4.4. Other Biologic Agents

#### 4.4.1. Rituximab

Rituximab is a monoclonal antibody that recognizes CD-20, an antibody expressed on the surface of mature B lymphocytes, approved for the treatment of moderate-to-severe RA [28]. It has been shown to be another promising agent in the treatment of refractory RA-associated scleritis, reported since 2009 [29]. In 2010, Iaccheri et al. reported a case of acute stromal keratitis and scleritis, with only partial response to cyclophosphamide, who was successfully controlled after two infusions of rituximab, off all steroids and without recurrence after 9 months of follow-up [25]. In the retrospective series of Fabiani et al., 4 out of 5 patients treated with rituximab improved by at least 2 step of scleritis grading system [30]. Adverse effects may include reactivation of viruses, as reported with acute retinal necrosis [31]. The rationale for the use of rituximab in necrotizing scleritis associated with RA is supported by immunohistochemical evidence, from analysis of enucleated eyes with necrotizing scleritis, suggesting that inflammation, in the group of eyes with association to a systemic autoimmune disease, may be driven by B cells, while macrophages could play a role in the necrotizing process [32]. A phase I/II dose ranging randomized trial [33] evaluated treatment with 2 infusions of either 500 mg or 1000 mg at day 1 and day 15, along with intravenous methylprednisolone 100 mg, and follow-up every 4 weeks for 24 weeks. In both dosing groups, rituximab was found effective, as in 9 out of 12 patients the inflammation was successfully controlled within 24 weeks of therapy. Peri-infusional disease exacerbations were noted in the 2- to 8-week period after infusion, requiring management with short term oral corticosteroid and careful tapering.

#### 4.4.2. Gevokizumab, Abatacept, and Tocilizumab

Gevokizumab is an anti-IL-1β monoclonal antibody. It has been reported to be effective in the control of refractory RA-associated scleritis [34]. A phase I/II open-label trial [35] included 8 patients treated for active non-infectious non-necrotizing scleritis with subcutaneous injection of 60 mg gevokizumab every 4 weeks for 12 weeks. Seven patients achieved a 2-step reduction of activity (on a 4-point scale according to standardized photographs) from baseline, with no serious adverse event.

The use of abatacept and tocilizumab, although quite common in RA, has only been reported in case reports and small retrospective case series. Tocilizumab was reported effective to control inflammation and steroid sparing in 50% of patients with refractory scleritis [36]. In the retrospective series of Fabiani et al. [30], 2 patients (3 eyes) were successfully treated with 162 mg/week of tocilizumab and 3 patients (4 eyes) treated for diffuse anterior scleritis, 2 patients (2 eyes) were successfully controlled with 125 mg weekly abatacept.

### 4.5. Other Perspectives

#### 4.5.1. Tofacitinib

Most recently, the JAK inhibitors are showing promising results in the control of RA inflammation. Tofacitinib is currently investigated for its efficacy in ocular inflammation (ClinicalTrials.gov (accessed on 11 May 2021) Identifier: NCT03580343). Its effectiveness in refractory scleritis has been suggested in case reports [30,37], including in patients with necrotizing scleritis [38].

#### 4.5.2. ACTH Gel

ACTH gel, a highly purified preparation of adrenocorticotropic hormone (ACTH) in a gel that is designed to provide extended release, is used in a variety of inflammatory diseases, including lupus, multiple sclerosis, or kidney diseases. In the treatment of nephrotic syndrome, ACTH gel is administered twice weekly by subcutaneous injections, for a course of treatment of 6 months or more, with approximately 25% of patients experiencing adverse effects similar to steroids [39]. Because it blocks multiple inflammatory pathways, ACTH gel could be beneficial in the treatment of scleritis, and it is currently under investigation in a phase II open trial, comparing subcutaneous injections of 80 units twice weekly versus thrice weekly (ClinicalTrials.gov (accessed on 11 May 2021) Identifier: NCT03465111).

### 4.6. Subconjunctival Injections: Triamcinolone, Sirolimus

Options for local treatments are limited, but studies have reported that subconjunctival injections of triamcinolone were able to provide resolution of symptoms and signs in 97% of treated eyes, 50% being recurrence-free 24 months after a single injection, and to reduce the need for oral NSAIDs, steroids, and immunosuppressants [40,41,42]. Significant side effects include cataract and ocular hypertension or glaucoma, the latter needing surgical intervention in a small proportion of patients [42].

One open-label prospective study [43] showed that a subconjunctival injection of sirolimus, a m-TOR inhibitor, 15 µL (660 µg) in the affected quadrant, was effective in reducing scleral inflammation by 8 weeks, though recurrences occurred requiring reinjection. A local sterile subconjunctival inflammation was noted after injection, but no ocular or systemic serious adverse event occurred.

Thus, subconjunctival injections of triamcinolone or sirolimus could be a valuable option for patients with poor compliance with eyedrops, but also patients with severe comorbidities not eligible to systemic treatment. However, local treatment should not be considered for patients with necrotizing scleritis. In addition to the risk of globe perforation, the onset of necrotizing scleritis is a strong indicator that the disease has transformed into a systemic microvasculitic disease, which needs to be addressed with vigorous systemic immunomodulation [44,45].

## 5. Necrotizing Scleritis and Rheumatoid Vasculitis

The onset of necrotizing scleritis may precede or be concurrent with systemic rheumatoid vasculitis. In a patient with RA, even when the disease seems quiescent or stable, the occurrence of necrotizing scleritis or, similarly, PUK, is an evidence of slowly emerging, potentially lethal, visceral vasculitis [45]. In 1976, 27% of patients with necrotizing scleritis were dead within 8 years [6]. Jones et al. [46] reported a mortality rate of over 36% in 3 years after a RA-associated scleritis and a 60% incidence of cardio-vascular events. Other reports from the pre-biologics era showed that up to 40% of patients with systemic rheumatoid vasculitis died within 5 years from systemic injuries of vasculitis, cardio-vascular events, or complications of treatments [47,48,49,50,51]. Even before biologics were available, in 1984 Foster et al. [44] showed that patients treated aggressively with cyclophosphamide or methotrexate had a favorable life and ocular prognosis compared to patients managed with oral NSAIDs and steroids. The overall incidence of rheumatoid vasculitis has decreased over the past decade, from earlier and more aggressive management of RA as well as from decreased rates of smoking [52,53]. There is no consensus yet as to what sequence of treatment is optimal for ocular manifestations of rheumatoid vasculitis such as necrotizing scleritis or PUK. After infectious causes have been carefully ruled out, including if any doubt a trial antiviral treatment, it seems reasonable to offer intravenous pulse of methylprednisolone, or at least oral steroids, followed by initiation or escalation of immunosuppressants with a low threshold for establishment of a biologic treatment. Agents of choice would be anti TNF alpha for their rapidity of action and safety profile, or rituximab [52,53]. With adequate management of the systemic disease, it seems that the onset of scleritis in a RA patient is not necessarily a life-threatening event anymore. In the recent study of Caimmi et al., even though 39% of episcleritis and 29% of scleritis patients had developed a new extra articular manifestation of RA within 5 years, the survival rate at 10 years was comparable between patients with inflammatory ocular disease and a comparation group of RA patients without ocular disease. Similarly, there was no difference in the incidence of cardio-vascular events between groups of RA patients with or without inflammatory ocular disease [3].

## 6. Complications and Incidence of Resolution

Complications can be seen in 57% of scleritis [3,6] and include decrease in visual acuity, keratitis, cataract, ocular hypertension and glaucoma [10], and scleral thinning and defects. Fortunately, most patients retain a good vision [3,10], unless they develop persistent structural damage to the eye, cystoid macular oedema, peripheral corneal melt, interstitial keratitis, or scleritis-induced astigmatism [7]. In the 1976 series of Watson and Hayreh, four patients received corneo-scleral grafts, two patients a scleral graft, three patients underwent enucleation of the affected eye [6].

The incidence of resolution of ocular disease at 1 year is only 40% in scleritis, as compared to 60% in episcleritis [3]. Patients with scleritis associated with a systemic disease were more likely to have inflammation ongoing for more than 5 years in the study of Bernauer et al. [7], and the presence of circulating antibodies (RF, ANA, and ANCA) tended to increase the risk of persistent inflammation. Remission, defined as no active inflammation for at least 3 months after discontinuing all immunosuppressive medication [54], was obtained in only 8% of patients with RA-associated scleritis, compared to 30% of patients without RA. When remission was obtained, 86% remained in remission after 1 year of follow-up [54].

## 7. Conclusions

Severe, refractory cases of scleritis are a rare manifestation of extra-articular RA. Early recognition and appropriate treatment are crucial and again, good cooperation with the rheumatology or internal medicine specialist will be key. Although no consensus or guidelines exist, many options issued from the rheumatology practice will become available for the treatment of refractory scleritis (Table 1), once infectious causes are ruled out, allowing a rapid control of the inflammation, and avoiding both structural damage to the eye and complications of long-term steroid use. Larger series and trials are needed to determine the best escalation strategy. However, the life-threatening prospect of rheumatoid vasculitis, following the onset of scleritis, already seems to have changed thanks to a more effective and safer control of the systemic inflammation.

## Figures and Tables

**Figure 1 jcm-10-02118-f001:**
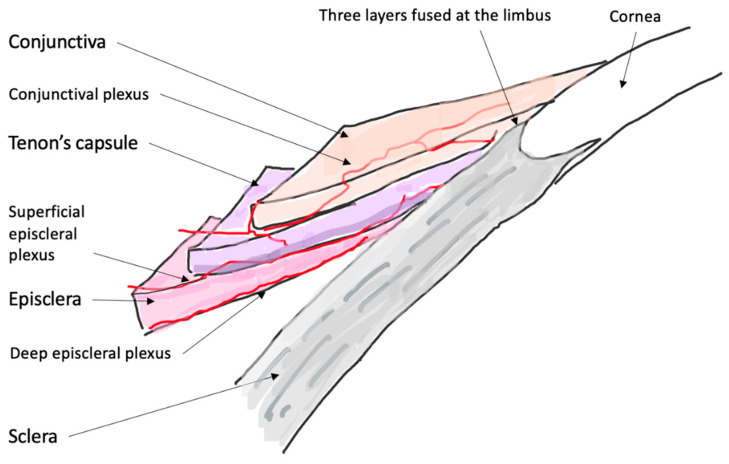
Cross sectional diagram of the normal anatomy of the outer layers of the eye. The most superficial layer is the conjunctiva with the conjunctival vascular plexus, easily moved over the underlying structures. Underneath, the Tenon’s capsule and the episclera, with an episcleral plexus of straight, radially arranged vessels. The deepest is the scleral plexus, tightly adherent to the sclera.

**Figure 2 jcm-10-02118-f002:**
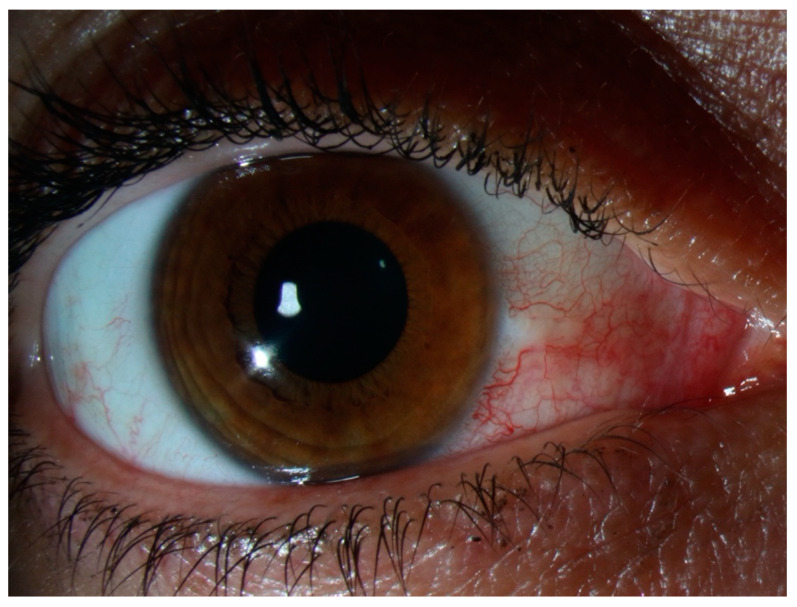
Episcleritis. There is no scleral swelling, and although the deep vascular plexus is congested, oedema and infiltration are limited to the episclera. The eye redness is most often limited to one sector of the eye, with a salmon pink to red discoloration. The pain is generally mild.

**Figure 3 jcm-10-02118-f003:**
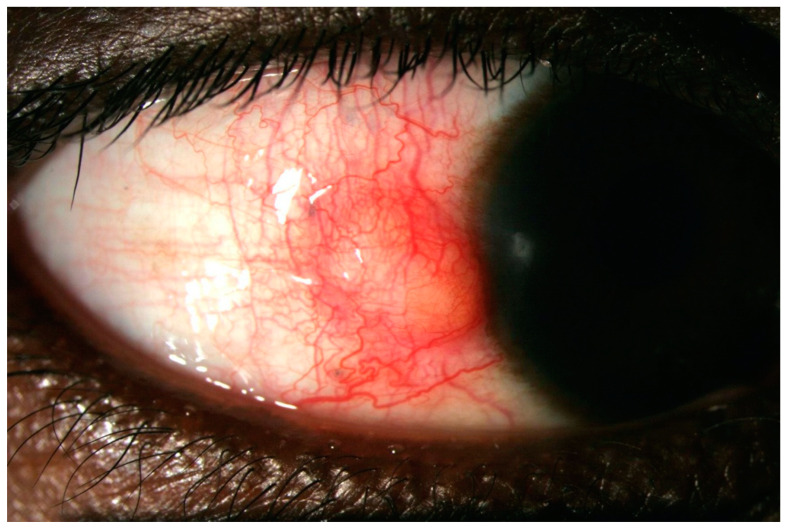
Nodular anterior scleritis. A scleral nodule, totally immobile, lifts the vessels. The nodule is clearly separated from the overlying congested episcleral tissue.

**Figure 4 jcm-10-02118-f004:**
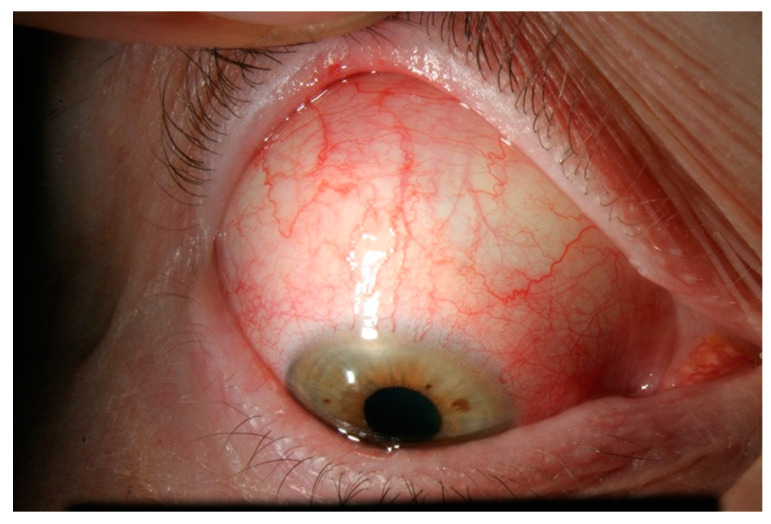
Diffuse anterior scleritis. Widespread inflammation of sclera, and abnormal appearance of the deep scleral vascular plexus. The scleral plexus becomes visible after blanching the superficial vessels with phenylephrine.

**Figure 5 jcm-10-02118-f005:**
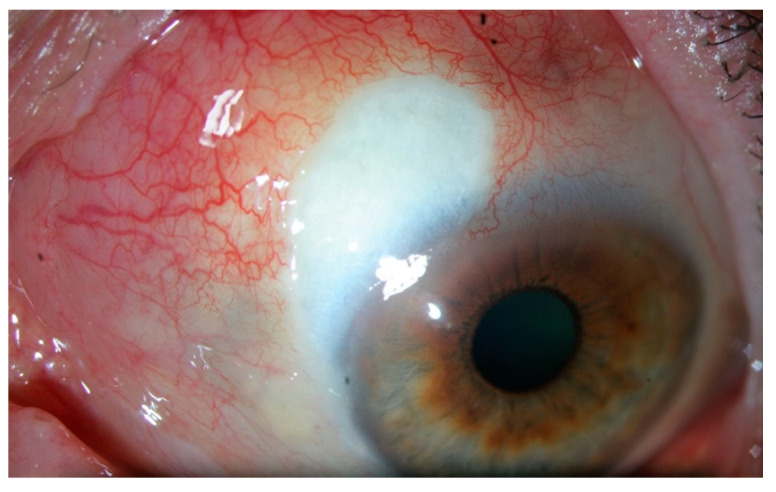
Necrotizing scleritis with inflammation. This form presents like a diffuse scleritis with an area of avascular, necrotic sclera.

**Figure 6 jcm-10-02118-f006:**
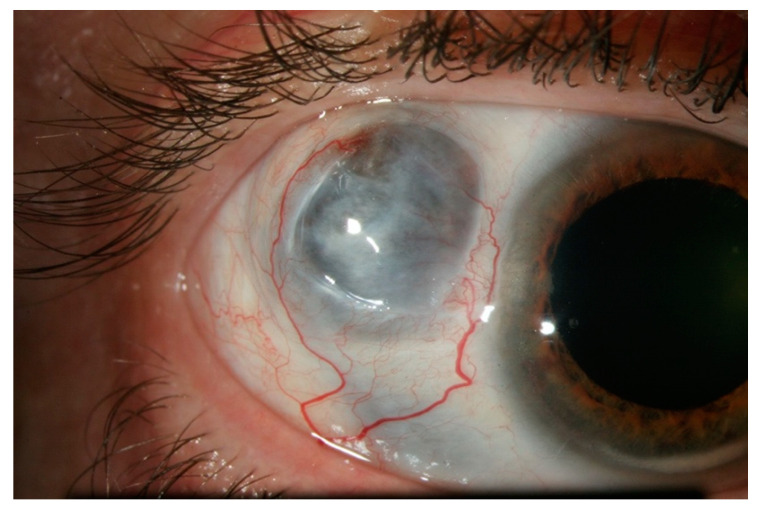
Necrotizing scleritis without inflammation, or scleromalacia perforans. This form presents with no inflammation and no pain, and almost complete destruction of scleral and episcleral tissue in one or more necrotic patch. The greyish appearance comes from the visible uveal tissue underneath.

**Figure 7 jcm-10-02118-f007:**
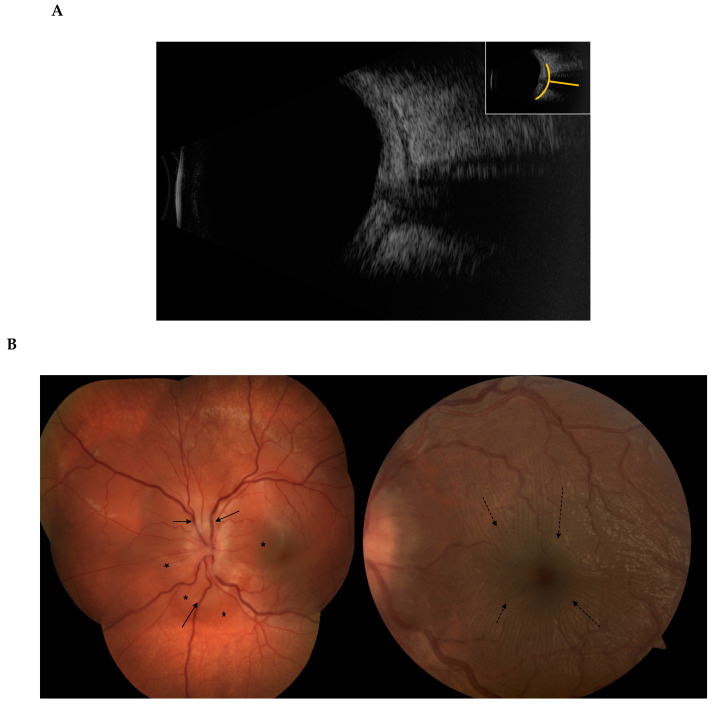
Posterior scleritis. The scleral inflammation involves the posterior part of the globe, with no apparent external sign unless there is an associated anterior component. (**A**) B-scan ultrasound showing the classic T-sign from scleral oedema, (**B**) The scleral inflammation can cause exudative retinal detachment (*), retinal folds (dashed arrows), retinal veins congestion (arrows) and optic disc swelling.

**Figure 8 jcm-10-02118-f008:**
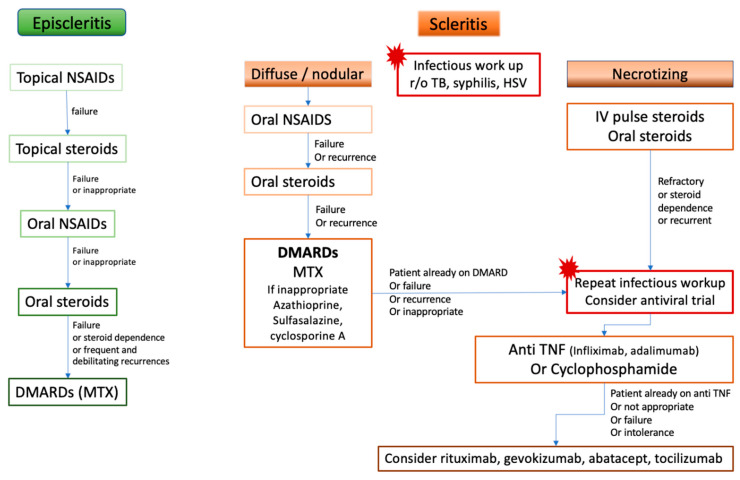
Proposition of treatment algorithm and strategy of escalation in rheumatoid arthritis-associated episcleritis and scleritis. DMARDs: disease-modifying anti-rheumatic drugs, HSV: Herpes Simplex Virus, IV: intravenous, MTX: methotrexate, NSAIDs: non-steroidal anti-inflammatory drugs, r/o: rule out, TB: tuberculosis, TNF: Tumor necrosis factor.

**Table 1 jcm-10-02118-t001:** New treatment perspectives: agent, level of available evidence, route of administration and dosage. AAO: American Academy of Ophthalmology, ACTH: adrenocorticotropic hormone, IV: intravenous, JAK: Janus Kinase, PO: per os, SC: subcutaneous, TNF: Tumor necrosis factor.

Class	Agent	Level of Evidence	Dose and Route of Administration
Biologics–Anti TNF-alpha	Infliximab	-retrospective series -phase I open trial -AAO recommendation “should be considered for patients with scleritis who have failed first-line immunomodulatory therapies”	IV 5 mg/kg week 0, 2, 6 and every 4 weeks
	Adalimumab	Retrospective series AAO recommendation “should be considered for patients with scleritis who have failed first-line immunomodulatory therapies”	SC 40 mg every 2 weeks
	Certolizumab pegol	Case reports	SC 400 mg week 0, 2 and 4, then 200 mg every 2 weeks or 400 mg every 4 weeks
Biologics-others	Rituximab	-Retrospective series -Phase I/II dose ranging randomized trial	IV 2g/6 months
	Gevokizumab	Phase I/II open label trial	SC 60 mg every 4 weeks
	Tocilizumab	Retrospective series	SC 162 mg/week
	Abatacept	Retrospective series	SC 125 mg weekly
JAK inhibitors	Tofacitinib	Case reports	PO 5 mg × 2/day
Others	ACTH gel	Ongoing prospective open trial	SC 80 units 2 or 3 times/week

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
