# Peer review of "Rheumatoid Arthritis Associated Episcleritis and Scleritis: An Update on Treatment Perspectives"

_jcm, 2021, doi:10.3390/jcm10102118_

Round 1

Reviewer 1 Report

Thank you for taking onboard my suggestions from my initial review. I would like to congratulate the authors on both the cross-sectional diagram of the layers of the eye, which really enhanced my understanding of their descriptions of pathology (being a non-ophthalmologist myself), and also for the treatment summary diagram and table, which really helped to pull the whole review together. Well done.

Author Response

Dear Reviewer,

Thank you for your review. Your comments really helped us put together a better manuscript, easily accessible to non-ophthalmologists and with practical points for the clinical practice.

Best regards

Reviewer 2 Report

This is a review manuscript of RA-associated episcleritis and scleritis. The topic of this manuscript is clinically relevant and interesting. I found the following points to be corrected in page 6.

Lines 107-108: GPA is a form of AAV.

LIne 110: GPA should be spelled out on the first appearance.

LIne 115: the word "immunopositive disease" is unclear. Seropositive or autoantibody-positive?

Author Response

Dear Reviewer, 

Thank you for reading our manuscript and allowing us to revise the following points according to your comments:

Lines 107-108: GPA is a form of AAV.

The sentence "patients with ANCA-associated vasculitis and granulomatosis with polyangiitis (GPA)" 

was revised to: "patients with ANCA-associated vasculitis such as granulomatosis with polyangiitis (GPA)"

LIne 110: GPA should be spelled out on the first appearance.

The paragraph was revised so that GPA is now spelled out in the first appearance line 108 "granulomatosis with polyangiitis".

LIne 115: the word "immunopositive disease" is unclear. Seropositive or autoantibody-positive?

Our rheumatology team prefers and advocates for the use of “immunopositive” instead of “seropositive disease” (please see: Vincent Goëb, The term "seropositive" to qualify polyarthritis/rheumatoid arthritis should be banned”. Joint Bone Spine 2018 Jan;85(1):123).

We changed in the manuscript "immunopositive" for "auto-antibody positive": "More than 80% of patients had an auto-antibody positive disease"

Thank you

Best regards